# Hepatocyte Specific gp130 Signalling Underlies APAP Induced Liver Injury

**DOI:** 10.3390/ijms23137089

**Published:** 2022-06-25

**Authors:** Jinrui Dong, Wei-Wen Lim, Shamini G. Shekeran, Jessie Tan, Sze Yun Lim, Joyce Wei Ting Goh, Benjamin L. George, Sebastian Schafer, Stuart A. Cook, Anissa A. Widjaja

**Affiliations:** 1Cardiovascular and Metabolic Disorders Program, Duke-National University of Singapore Medical School, Singapore 169857, Singapore; djrnku@gmail.com (J.D.); lim.wei.wen@nhcs.com.sg (W.-W.L.); shamini_g@duke-nus.edu.sg (S.G.S.); sy76@nus.edu.sg (S.Y.L.); joyce.goh@duke-nus.edu.sg (J.W.T.G.); ben.george@duke-nus.edu.sg (B.L.G.); sebastian@duke-nus.edu.sg (S.S.); 2National Heart Research Institute Singapore, National Heart Centre Singapore, Singapore 169857, Singapore; gmstje@nus.edu.sg; 3MRC-London Institute of Medical Sciences, Hammersmith Hospital Campus, London W12 0NN, UK

**Keywords:** IL-11, IL-6, GP130, IL6ST, liver injury, hepatotoxicity, acetaminophen, paracetamol, liver damage

## Abstract

N-acetyl-p-aminophenol (APAP)-induced liver damage is associated with upregulation of Interleukin-11 (IL11), which is thought to stimulate *IL6ST* (gp130)-mediated STAT3 activity in hepatocytes, as a compensatory response. However, recent studies have found IL11/IL11RA/gp130 signaling to be hepatotoxic. To investigate further the role of IL11 and gp130 in APAP liver injury, we generated two new mouse strains with conditional knockout (CKO) of either *Il11* (CKO*^Il11^*) or gp130 (CKO^gp130^) in adult hepatocytes. Following APAP, as compared to controls, CKO^gp130^ mice had lesser liver damage with lower serum Alanine Transaminase (ALT) and Aspartate Aminotransferase (AST), greatly reduced serum IL11 levels (90% lower), and lesser centrilobular necrosis. Livers from APAP-injured CKO^gp130^ mice had lesser ERK, JNK, NOX4 activation and increased markers of regeneration (PCNA, Cyclin D1, Ki67). Experiments were repeated in CKO*^Il11^* mice that, as compared to wild-type mice, had lower APAP-induced ALT/AST, reduced centrilobular necrosis and undetectable IL11 in serum. As seen with CKO^gp130^ mice, APAP-treated CKO*^Il11^* mice had lesser ERK/JNK/NOX4 activation and greater features of regeneration. Both CKO^gp130^ and CKO*^Il11^* mice had normal APAP metabolism. After APAP, CKO^gp130^ and CKO*^Il11^* mice had reduced *Il6*, *Ccl2*, *Ccl5*, *Il1β*, and *Tnfα* expression. These studies exclude IL11 upregulation as compensatory and establish autocrine, self-amplifying, gp130-dependent IL11 secretion from damaged hepatocytes as toxic and anti-regenerative.

## 1. Introduction

The Interleukin-6 (IL6) signal transducer (*IL6ST*/gp130) is an important cell surface receptor [1]. During evolution, signaling via gp130 has acquired complexity with multiple IL6 family members binding directly to gp130 or indirectly to gp130 via their cognate alpha receptors. While gp130 itself is widely expressed, IL6 family member alpha receptors have variable expression with IL11RA and IL6R expressed at different levels on divergent cell types [2,3]. The major pathways activated downstream of gp130 are JAK/STAT3 and Ras/ERK, with Akt activation also featured. 

The role of gp130 in the liver has been the subject of much investigation. The literature mostly reports that gp130-mediated signaling is protective across acute and chronic mouse models of liver injury [4,5,6,7,8]. It is widely accepted that IL6-driven STAT3 activity underlies gp130-mediated hepatoprotection in the injured liver: a synthetic fusion protein of IL6 and its receptor (HyperIL6) promotes hepatic regeneration in mice subjected to d-galactosamine administration [9,10]. However, although HyperIL6 is reproducibly hepatoprotective, its effects now appear unrelated to STAT3 activity [11].

Interleukin-11 (IL11) is a member of the IL6 family cytokine that has pathogenic activity in the liver [12,13]. However, IL11-stimulated STAT3 activity is also suggested to protect the liver: an injection of recombinant human IL11 (rhIL11) into mice reduces Concanavalin A-induced, T-cell-mediated hepatotoxicity and N-acetyl-p-aminophenol (APAP)-induced liver damage [14,15]. The beneficial effects of IL11 in the liver have been studied in the most detail in the context of APAP injury where its high levels of secretion from injured hepatocytes is thought to be compensatory [16,17]. More recent experiments have challenged the earlier findings as species-matched IL11 was shown to cause hepatocyte dysfunction and liver inflammation [18,19,20]. Some of these disparities may reflect the use of rhIL11 in the earlier studies, as rhIL11 was recently shown not to faithfully activate the mouse IL11RA receptor [18].

Mice globally deleted for gp130 die at embryonic day 12.5 post-conception with placental disruption and global hypoplastic organ development [21]. Hepatocyte-specific gp130 knockouts have been generated by crossing albumin promoter/AFP enhancer Cre mice (Alfp-Cre; [22]) to mice with floxed gp130 alleles. In these mice, the deletion of gp130 from day 10.5 post-conception is associated with greater liver damage following an injection of lipopolysaccharide [7]. However, we and others have found that deletion of *Il11ra1* during development is maladaptive in the liver whereas deletion of *Il11ra1* in hepatocytes in the adult is protective [16,18]. It remains an open question if global inhibition of gp130 signaling in the adult hepatocytes is detrimental or if its deletion during development primes the liver for damage in adulthood.

In the current study, we set out to address the following questions: (1) is gp130 signaling in hepatocytes protective or injurious in the context of APAP-induced liver injury; (2) are hepatocytes a major source of IL11 secretion following APAP injury; and (3) is endogenous IL11 activity in APAP-injured hepatocytes adaptive or maladaptive? To do so, we generated two new transgenic mouse models: mice with loxP-flanked gp130 (gp130^loxP/loxP^) and mice with loxP-flanked *Il11* (*Il11^loxP/loxP^*). Mice were injected with AAV8 expressing Cre under the control of the albumin promoter to create conditional knockout (CKO) mice deleted for the floxed alleles (CKO^gp130^ or CKO*^Il11^*) in adult hepatocytes and these mice were then subjected to APAP injury. 

## 2. Results

### 2.1. Hepatocyte-Specific gp130-Deficient Mice

We introduced loxP sites into the *Il6st*/gp130 gene locus using the CRISPR/Cas9 system for the conditional deletion of exons 4–5 (Figure 1A). Mice homozygous for loxP-flanked gp130 alleles (gp130^loxP/loxP^) were generated on a C57BL/6N background and genotypes were determined by sequencing and PCR (Figure 1B). To conditionally delete gp130 in hepatocytes (CKO^gp130^), gp130^loxP/loxP^ mice were infected with either AAV8 encoding Cre under the control of the albumin promoter (AAV8-*Alb*-iCre) or AAV8-*Alb*-Null to generate CKO^gp130^ and control mice, respectively (Figure 1C). While albumin-driven Cre expression from germline in the mouse is associated with Cre protein in cells beyond hepatocytes [23], AAV8-Cre, when combined with an albumin promoter, has hepatocyte-restricted expression in adults, enabling conditional and temporal gene regulation in hepatocytes [24].

The heart, liver, lung, and kidney were harvested 3 weeks post-infection and qPCR analysis showed notable downregulation of gp130 transcripts in the liver (−85%, *p* < 0.0001), whereas gp130 mRNA levels were unaltered in the heart, lung, or kidney (Figure 1D). The liver-specific effect in the CKO^gp130^ mice was confirmed by immunoblotting with reduced gp130 protein levels in the liver (89%, *p* < 0.0001), but with normal levels of expression in the heart, lung, and kidney (Figure 1E,F and Appendix A). Immunohistochemistry (IHC) staining further confirmed gp130 deletion in hepatocytes of CKO^gp130^ mice as compared to controls (Figure 1G). By serum biochemistry, immunoblotting, gene expression, and histological analysis, deletion of gp130 for 3 weeks in adult hepatocytes had no measurable effect (Figure 2 and Figure 3). These data describe a new mouse model with a conditional knockout of gp130 in adult mouse hepatocytes: the CKO^gp130^ mouse.

### 2.2. Effects of gp130 Deletion in Adult Hepatocytes on Liver Injury and Regeneration

To ensure that livers of CKO^gp130^ mice were able to metabolize APAP normally, we probed for the expression of hepatic cytochrome P450 2E1 enzyme (CYP2E1), a key enzyme required for the conversion of APAP into its reactive and cytotoxic metabolite, N-acetyl-p-benzoquinone imine (NAPQI), which rapidly depletes glutathione (GSH) levels (Figure 2A). Immunohistochemistry revealed that livers from both control and CKO^gp130^ groups had a similar centrilobular expression of hepatic CYP2E1 levels (Figure 2A). To determine if the effects of gp130 deletion in adult hepatocytes differs from its deletion during early development, which is detrimental [7,8], we subjected both CKO^gp130^ mice and wild-type littermates to drug-induced liver injury using APAP (Figure 2B).

Consistent with the similar CYP2E1 expression observed in both genotypes (Figure 2A), GSH concentrations were equally depleted in CKO^gp130^ and WT mice at 0.5h post-APAP dosing, which reflects normal APAP metabolism and NAPQI generation in both strains (Figure 2C). By 6 h post-APAP, livers of CKO^gp130^ mice had begun to restore GSH levels as compared to wild-type littermate controls (Figure 2C). As compared to WT controls, serum markers of liver damage (ALT (alanine transaminase) and AST (aspartate aminotransferase)) were lower in CKO^gp130^ mice 6 h post-APAP, consistent with reduced acute hepatocyte injury (Figure 2D,E).

We have shown previously that deletion of *Il11ra1* in adult hepatocytes protects the mouse liver from APAP-induced hepatotoxicity, despite normal APAP metabolism and NAPQI generation, which is associated with reduced ERK, JNK, and NOX4 activation [18]. We therefore examined the expression of IL11 and its reported downstream signaling molecules, i.e., NOX4, ERK, JNK, Caspase3, as well as STAT3 in CKO^gp130^ and control mice, and found increased levels of all of these factors in the livers of control mice 6 h following APAP administration (Figure 2F and Appendix A). The expression levels of these molecules were significantly reduced in the CKO^gp130^ mice, except for STAT3, whose expression was increased as compared to those of wild-type control mice (Figure 2F and Appendix A). We highlight that caspase 3 was used as a generic marker of cellular damage/death and that pharmacologic inhibition of caspases does not protect the liver from APAP-induced damage [25,26]. As compared to APAP-injured control mice, CKO^gp130^ mice also had lesser centrilobular necrosis, the pathognomonic histological feature of APAP liver injury (Figure 2G).

We performed a second set of experiments at 24 h following APAP dosing to further assess the extent of liver damage and early liver regeneration phenotypes. Consistent with findings from previous studies, serum IL11 levels were highly elevated (saline, 0 ng/mL (not detected by the assay); APAP, 3 ng/mL) in wild-type mice at this time point [16,18,19] (Figure 3A). In contrast, IL11 levels in CKO^gp130^ mice post-APAP were ~90% lower (*p* < 0.0001), as compared to wild-type mice (Figure 3A). The lower levels of IL11 in the serum of CKO^gp130^ mice were associated with reductions in ALT and AST levels (reduction (%): ALT, 86.6; AST, 72.6) and higher hepatocyte GSH levels, which were comparable to the levels observed in saline-injected mice (Figure 3B–D). 

Immunoblotting confirmed reduced IL11 protein levels in livers of CKO^gp130^ mice, placing IL11 secretion downstream of gp130 signaling in the hepatocytes of APAP-injured liver (Figure 3E and Appendix A). CKO^gp130^ mice exhibited increased expression of PCNA and Cyclin D1, consistent with an early signal of enhanced regeneration post-APAP injury (Figure 3E and Appendix A). There was a consistent reduction in the expression of a range of pro-inflammatory markers (*Ccl2*, *Ccl5*, *Il1β*, *Il6*, and *Tnfα*) in the APAP-injured CKO^gp130^ mice, as compared to wild-type controls (Figure 3F).

At the signaling level, APAP-injured CKO^gp130^ livers, which are protected from APAP damage, had lesser activation of ERK, JNK, and Caspase 3, as compared to controls (Figure 3G and Appendix A). In contrast to the upregulation of pSTAT3 observed in APAP-dosed CKO^gp130^ mice at a 6 h time point, the hepatic levels of STAT3 activation on the day after APAP (24 h) were similar to those of the saline-injected mice (Figure 3G and Appendix A).

As compared to APAP-injured control mice, CKO^gp130^ mice had lesser centrilobular necrosis (Figure 3H). While this may reflect reduced early liver damage (ALT, AST), it could also be due to greater and/or faster hepatocyte regeneration, as suggested by the elevated PCNA and Cyclin D1 levels (Figure 3E and Appendix A), or a combination of both phenomena. Immunostaining for Ki67, a proliferation marker, revealed a much larger degree of hepatocyte proliferation consistent with enhanced hepatocyte regeneration in the centrilobular regions of injured CKO^gp130^ livers, as compared to controls (Figure 3I).

Taken together, these data show that genetic deletion of gp130 in hepatocytes: (1) does not impact APAP metabolism or GSH depletion; (2) confers hepatoprotective effects as early as 6 h post-APAP toxicity; and (3) promotes a pro-regenerative state in the liver 24 h following APAP injury. 

### 2.3. Hepatocyte-Specific Il11-Deficient Mice

The source of IL11 secretion from the liver following APAP- or dietary-induced liver injury remains a point of discussion concerning the hepatocytes [16,17,18,19] and hepatic stellate cells both implicated [3]. To define the primary source of IL11 following APAP injury, we generated mice with hepatocyte-specific deletion of *Il11*.

Mice deleted for *Il11* in hepatocytes were created by introducing a targeting vector with homology arms for the *Il11* gene containing loxP sites flanking the coding sequence of exons 2–5 with an adjoining rBG-polyA sequence, a neomycin-resistant cassette flanked by self-deletion anchor sites, as well as a separate construct comprising the endogenous splice acceptor of intron 1, a 2A-EGFP cassette, and the specialized termination sequence (BGH-polyA) into the intron 1 of the *Il11* gene (Figure 4A). Mice homozygous for loxP-flanked *Il11* alleles (*Il11^loxP/loxP^*) were generated on a C57BL/6 background and genotypes were determined by sequencing and PCR (Figure 4B).

Wild-type or *Il11^loxP/loxP^* mice were infected with AAV8-*Alb*-Cre or AAV8-*Alb*-Null and their organs were harvested 3 weeks post-infection (Figure 4C). There was a significant downregulation of hepatic *Il11* mRNA and IL11 protein in the CKO*^Il11^* mice (mRNA: 89%; protein: 76% lower than control mice), whereas both mRNA and protein levels were unaltered in the heart, lung, or kidney (Figure 4D,E and Appendix A). *EGFP* levels were increased in the liver, consistent with Cre-mediated recombination at the targeted locus, and *EGFP* was not detected in other tissues, further confirming the conditionality of gene targeting (Appendix A). Deletion of *Il11* for 3 weeks in adult hepatocytes at baseline had no effect on serum biochemistry, gene expression, or histology (Figure 5 and Figure 6). These studies describe a new mouse model with a conditional knockout of *Il11* in adult hepatocytes: the CKO*^Il11^* mouse.

### 2.4. Effects of Il11 Deletion in Adult Hepatocytes on Liver Injury and Regeneration

We determined if the conditional deletion of *Il11* in adult hepatocytes (CKO*^Il11^*) was protective or damaging in the context of APAP-induced liver injury. Despite having normal CYP2E1 expression and a similar extent of acute glutathione depletion (0.5 h post-APAP), CKO*^Il11^* mice had lower serum markers of hepatocyte damage (ALT and AST), higher hepatic GSH concentrations, and lesser extent of centrilobular necrosis by 6 h post-APAP, as compared to wild-type controls (Figure 5A–E,G). Following APAP, IL11 levels were elevated in the wild-type controls but undetected in CKO*^Il11^* (Figure 5F and Appendix A). Signaling changes largely replicated those observed in CKO^gp130^ mice with notable attenuation of NOX4 expression and ERK, JNK, and Caspase 3 activation in APAP-dosed CKO*^Il11^* mice, as compared to injured control mice (Figure 5F and Appendix A). In contrast, hepatic STAT3 activation at 6 h post-APAP was unchanged by the loss of IL11 signaling in hepatocytes.

One day after APAP (24 h), IL11 levels were detected at very high levels in the serum of wild-type mice (3.3 ± 0.6 ng/mL) but undetectable in CKO*^Il11^* mice (Figure 6A). Immunoblotting revealed reduced IL11 in livers of CKO*^Il11^* mice, as compared to wild-type controls (Figure 6B and Appendix A). Thus, following APAP injury, IL11 is primarily secreted from injured hepatocytes and detectable locally in the liver and also in the serum. As compared to wild-type mice, liver damage markers were markedly reduced in CKO*^Il11^* mice (ALT: 46.4% lower and AST: 40.0% lower) along with increased hepatocyte GSH levels (Figure 6C–E). We noted that the reduction in ALT and AST post-APAP was less pronounced in CKO*^Il11^* mice at both the 6 and 24 h time points when compared to CKO^gp130^ animals (Figure 2C–E, Figure 3B–D, Figure 5C–E and Figure 6B–D).

As compared to controls, CKO*^Il11^* mice had increased PCNA and Cyclin D1 levels 24 h following APAP-induced liver damage, consistent with greater hepatic regeneration in the injured liver, as well as reduced pro-inflammatory gene expression (*Ccl2*, *Ccl5*, *Il1β*, *Il6*, and *Tnfα*) (Figure 6B,F and Appendix A). As seen in the published data from CKO*^Il11ra1^* mice [18] and the CKO^gp130^ mice data above (Figure 3G), there was lesser ERK and JNK activation and Caspase 3 cleavage in livers of CKO*^Il11^* mice post-APAP, as compared to controls (Figure 6G and Appendix A). In contrast, there was no increase in hepatic pSTAT3 following APAP in wild-type mice in this set of experiments (Figure 6G and Appendix A).

In histological studies, CKO*^Il11^*mice exhibited lesser centrilobular necrosis and had greater hepatic Ki67 staining, as compared to wild-type controls (Figure 6H,I). These data are concordant with the studies in the CKO^gp130^ mice, although the degree of liver damage in CKO*^Il11^*mice by histology was slightly greater and the number of Ki67 positive cells was slightly fewer than in CKO^gp130^ mice (Figure 3H,I).

## 3. Discussion

IL11 was originally thought to protect the liver and promote regeneration during APAP-induced liver injury [14,16] and other forms of liver damage [9,10]. However, recent studies have questioned earlier assumptions as IL11 appears to cause hepatic fibrosis, steatosis, and inflammation in a number of liver diseases [3,19,20]. IL6 signaling is also believed to be protective in the liver, perhaps relating to IL6 *trans*-signaling [5], but the specificity of this effect was recently questioned [11]. Here, we attempted to address these discrepancies by deleting either gp130 or IL11 in hepatocytes in adult mice that were then subjected to APAP injury. 

IL6 is elevated in the serum 24 h after APAP injury [27] but only mildly increased in the liver itself [28]. We find here that STAT3 activation in APAP-injured livers is seen early on (6 h post-injury) but is variable across genotypes at 24 h post-injury and unrelated to liver injury/regeneration. Based on these data, endogenous IL6-mediated STAT3 activity (*cis* or *trans*) in hepatocytes seems excluded as being hepatoprotective in the APAP-injured liver. A beneficial role for IL6 in liver regeneration remains possible, indeed likely, through its activity in immune cells that are important for the removal of damaged hepatocytes to promote tissue damage resolution [29].

The published literature robustly demonstrates that the deletion of gp130 in hepatocytes during development is detrimental [8,30,31]. In addition, the deletion of gp130 in all cells in the adult using an interferon-inducible promoter is associated with delayed liver regeneration [6]. Other genetic models have been made to inhibit either gp130-dependent Ras/MEK/ERK or JAK/STAT activation in hepatocytes [30,31]. In these strains, mice lacking STAT3 activation were susceptible to disease whereas those lacking Ras/MEK/ERK were protected from liver damage. 

We show here that aggregate gp130 signaling (i.e., the integration of the effects of all ligands acting via gp130) in injured adult hepatocytes promotes APAP-induced liver damage and inhibits liver regeneration. The deletion of gp130 mirrored the effects seen with *Il11* deletion and inhibited ERK activation, which is IL11-dependent [11,18,19]. Thus, maladaptive IL11 signaling appears to be the dominant gp130-mediated effect in the APAP damaged liver. This is in keeping with data seen in mice with global and constitutive deletion of *Il11* (global IL11 KO) and in adult mice with hepatocyte-specific deletion of *Il11ra1* but not in mice with global and constitutive *Il11ra1* deletion [16,18].

Following APAP dosing of wild-type mice, serum IL11 levels increased from undetectable to ~3ng/mL, in the two experiments shown here. Serum IL11 was detected at much lower levels following APAP in CKO^gp130^ mice (0.31 ± 0.14 ng/mL) and was undetectable in CKO*^Il11^* mice. Our interpretation of this is that damaged CKO^gp130^ hepatocytes secrete some IL11 but lack the autocrine feed-forward signaling, which limits IL11 secretion, prevents additional cell death, and is permissive for hepatic regeneration. The CKO*^Il11^* data establish hepatocytes as the primary source of IL11 in the APAP injured liver and reinforce IL11 as a hepatotoxin.

This study resolves some questions about IL11 and gp130 but also stimulates new queries. For instance, why are CKO^gp130^ mice more strongly protected from APAP toxicity as compared to the CKO*^Il11^* mice? Additionally, the deletion of gp130 in adult hepatocytes has different effects compared to its deletion during development but the underlying mechanisms are unexplained. What are the relative contributions of cytoprotection versus regeneration in CKO^gp130^ and CKO*^Il11^* mice to liver health, given the interrelation of these phenomena? These unresolved issues require further study.

In the introduction, we posed three questions that we wished to address, which we review and answer here based on our findings: (1) *is gp130 signaling in hepatocytes protective or injurious in the context of APAP-induced liver injury?* We conclude that aggregate gp130 signaling is damaging. (2) *Are hepatocytes a major source of IL11 secretion following APAP injury?* Our data show that hepatocytes are the major source of IL11 secretion following APAP injury. (3) *Is endogenous IL11 activity in APAP-injured hepatocytes adaptive or maladaptive?* Our experiments, building on previous findings [11,18], show that endogenous IL11 upregulation in the damaged liver is toxic and anti-regenerative.

In conclusion, IL11 is secreted from APAP-injured hepatocytes to drive an autocrine, self-amplification loop of IL11-driven IL11 secretion, leading to hepatocyte death via gp130-mediated ERK-related signaling. While earlier publications suggested IL11 upregulation following APAP to be adaptive, this appears not to be the case. A meaningful protective role for endogenous IL6 *cis*- or *trans*-signaling in hepatocytes in the APAP liver injury appears excluded. To aid future research in this area, the mouse models described here are made available via the Jackson laboratory repository.

## 4. Materials and Methods

### 4.1. Ethics Statements

Animal studies were carried out in compliance with the recommendations in the Guidelines on the Care and Use of Animals for Scientific Purposes of the National Advisory Committee for Laboratory Animal Research (NACLAR). All experimental procedures were approved (SHS/2019/1482 and SHS/2019/1483) and conducted in accordance with the SingHealth Institutional Animal Care and Use Committee (IACUC).

### 4.2. Antibodies

Cleaved Caspase 3 (9664, Cell Signaling Technology (CST), Danvers, MA, USA), Caspase 3 (9662, CST, MA, USA), Cyclin D1 (55506, CST, MA, USA), phospho-ERK1/2 (4370, CST, MA, USA), ERK1/2 (4695, CST, MA, USA), GAPDH (2118, CST, MA, USA), gp130 (ab202850, Abcam, UK), IL11 (X203, Aldevron, Germany), p-JNK (4668, CST, MA, USA), JNK (9252, CST, MA, USA), Ki67 (ab16667, Abcam, UK), PCNA (13110, CST, MA, USA), p-STAT3 (4113, CST, MA, USA), STAT3 (4904, CST, MA, USA), anti-mouse HRP (7076, CST, MA, USA), anti-rabbit HRP (7074, CST, MA, USA).

### 4.3. Animal Models

All mice were housed in temperatures of 21–24 °C with 40–70% humidity on a 12 h light/12 h dark cycle and provided food and water ad libitum, except in the fasting period, during which only water was provided ad libitum.

#### 4.3.1. Gp130-Floxed Mice 

CRISPR/Cas9 was used to introduce loxP sequences into the mouse *Il6st*/gp130 locus (ENSMUSG00000021756) for the conditional deletion of exons 4–5, resulting in a null allele upon Cre recombinase-mediated excision. Cas9, gRNA (with recognition sites on introns 3 and 5), and the targeting construct containing two loxP sequences were co-injected into fertilized eggs for mutant mice production (Cyagen Biosciences Inc, Jiangsu, China). Insertion of loxP sites into the gp130 gene locus was verified by sequencing. Mutant gp130-floxed offspring were generated and maintained on a C57BL/6N background and identified by genotyping to detect the insertion of loxP sites using the following primers: 5′-TGGCTTTTAGGGCTAGAGAGAAGG-3′ (forward) and 5′-GATTTCCCTCAGGAAACAGACTGAG-3′ (reverse). Wild-type (WT) alleles were identified by a 138 bp PCR product while mutant alleles were identified by a 205 bp PCR product.

#### 4.3.2. Il11-Floxed Mice

The mouse *Il11* gene (ENSMUSG00000004371) consists of 5 exons, with an ATG start codon in exon 1 and a TGA stop codon in exon 5. A targeting vector with homology arms for *Il11* gene containing 3 constructs: (1) loxP sites flanking the coding sequence (CDS) of exons 2–5 and an adjoining rBG-polyA sequence, (2) a neomycin-resistant cassette flanked by self-deletion anchor sites, followed by (3) a separate construct that comprises the endogenous splice acceptor of intron 1, a 2A-EGFP cassette, and the specialized termination sequence (BGH-polyA), which were inserted into intron 1 of the *Il11* gene (Cyagen Biosciences Inc, China). Using this targeting strategy, normal *Il11* gene expression will be maintained by the exons 2–5 CDS in construct 1 in the non Cre-mediated excised gene (control). Following Cre recombination, *Il11* gene inactivation will result from the deletion of loxP-flanked exons 2–5 sequences in construct 1 and the subsequent expression of the termination sequence BGH-polyA. C57BL/6 ES cells were used for gene targeting and successful target clones were injected in C57BL/6 albino embryos, which were re-implanted into CD-1 pseudo-pregnant females. Founder animals were identified by their coat colour and their germline transmission was confirmed by breeding with C57BL/6 females and subsequent genotyping of the offspring. Mutant *Il11*-floxed offspring were identified by genotyping to detect the insertion of loxP sites in intron 1 using the following primers: 5′-TCCTCCCTCAGATCCAGGCGTTC-3′ (forward) and 5′-CTAGGGTTCGGAATTTTGGTCTTG-3′ (reverse). WT alleles were identified by a 239 bp PCR product while mutant alleles were identified by a 273 bp PCR product.

#### 4.3.3. Generation of Hepatocyte-Specific CKO^gp130^ or CKO*^Il11^* Mice

All AAV8 vectors used in this study were synthesized by Vector Biolabs, PA, USA and have been described previously [18,19]. To specifically delete gp130 *or Il11* in albumin-expressing cells, a AAV8-*Alb*-iCre vector was injected to mice homozygous for loxP-flanked gp130 alleles (gp130^loxP/loxP^) or loxP-flanked *Il11* alleles (*Il11^loxP/loxP^*) to create CKO^gp130^ or CKO*^Il11^* mice. AAV8-*Alb*-Null vector was used as a KI vector control.

#### 4.3.4. Mouse Models of Acetaminophen (APAP)

Using a model previously established in our laboratory [11,18], 9–11-week-old male CKO^gp130^, CKO*^Il11^*, or WT control mice were fasted overnight prior to intraperitoneal (IP) administration of APAP (400 mg/kg; A3035, Sigma, St. Louis, MO, USA). Mice were euthanized 0.5, 6, or 24 h post-APAP.

### 4.4. RT-qPCR

Total RNA was extracted from snap-frozen liver tissues using Trizol (Thermo Fisher Scientific, Carlsbad, CA, USA) and a RNeasy Mini Kit (Qiagen, Germantown, MD, USA). PCR amplifications were performed using an iScript cDNA Synthesis Kit (Biorad, Watford, UK). Gene expression was analyzed in duplicate by SYBR green (Qiagen) technology using StepOnePlus (Applied Biosystem, Waltham, MA, USA) over 40 cycles. Expression data were normalized to *Gapdh* mRNA expression and the fold change was calculated using the 2^−∆∆Ct^ method. Primer sequences are as follows: *Ccl2*, 5′-GAAGGAATGGGTCCAGACAT-3′ and 5′-ACGGGTCAACTTCACATTCA-3′; *Ccl5*, 5′-GCTGCTTTGCCTACCTCTCC-3′ and 5′-TCGAGTGACAAACACGACTGC-3′; *EGFP*, 5′-GGACGACGGCAACTACAAGA-3′ and 5′-AAGTCGATGCCCTTCAGCTC-3′; *Gapdh*, 5′-CTGGAAAGCTGTGGCGTGAT-3′ and 5′-GACGGACACATTGGGGGTAG-3′; *Il1β,* 5′-CACAGCAGCACATCAACAAG-3′ and 5′-GTGCTCATGTCCTCATCCTG-3′; *Il6*, 5′-CTCTGGGAAATCGTGGAAAT-3′ and 5′-CCAGTTTGGTAGCATCCATC-3′; *Il6st*/gp130, 5′-GCCAGCTACATCGTGTGGAA-3′ and 5′-TGACACTGGACGTGGTTCTG-3′; *Il11*, 5′-AATTCCCAGCTGACGGAGATCACA-3′ and 5′-TCTACTCGAAGCCTTGTCAGCACA-3′; *Tnfα*, 5′-ATGAGAAGTTCCCAAATGGC-3′ and 5′-CTCCACTTGGTGGTTTGCTA-3′.

### 4.5. Immunoblotting

Western blots were carried out from hepatocyte and liver tissue lysates. Hepatocytes and tissues were homogenized in a radioimmunoprecipitation assay (RIPA) buffer containing protease and phosphatase inhibitors (Thermo Fisher Scientific, Carlsbad, CA, USA), followed by centrifugation to clear the lysate. Protein concentrations were determined by Bradford assay (Bio-Rad, Watford, UK). Equal amounts of protein lysates were separated by SDS-PAGE, transferred to a PVDF membrane, and subjected to immunoblot analysis for the indicated primary antibodies. Proteins were visualized using the ECL detection system (Thermo Fisher Scientific, Carlsbad, CA, USA) with the appropriate secondary antibodies. 

#### Enzyme-Linked Immunosorbent Assay (ELISA) and Colorimetric Assays

The levels of IL11 in mouse serum were quantified using Mouse IL11 DuoSet ELISA (DY418, R&D Systems, Minneapolis, MN, USA). The levels of alanine transaminase (ALT) or aspartate aminotransferase (AST) in mouse serum and hepatocyte supernatant were measured using ALT Activity (ab105134, Abcam, Cambridge, UK) or AST (ab105135, Abcam, Cambridge, UK) Assay Kits. Liver glutathione sulfhydryl (GSH) measurements were performed using the Glutathione Colorimetric Detection Kit (EIAGSHC, Thermo Fisher Scientific, Carlsbad, CA, USA). All ELISA and colorimetric assays were performed according to the manufacturer’s protocol.

### 4.6. Histology

#### 4.6.1. Hematoxylin and Eosin (H&E) Staining

Livers were fixed for 48 h at room temperature (RT) in 10% neutral-buffered formalin (NBF), dehydrated, embedded in paraffin blocks, and sectioned at 7 μm. Sections were stained with H&E according to standard protocol and examined by light microscopy. Quantification of liver necrosis area was performed using ImageJ Fiji (version 1.50, National Institute of Health, Bethesda, MD, USA) by measuring H&E-negative areas (areas inside the red lines) relative to the tissue area (area of the whole image excluding the areas inside the yellow line) from 3 randomly selected 100× field images from each liver. A figure exemplifying the quantification analysis is provided below (Figure 7). ROI: Region of interest.

#### 4.6.2. Immuno-Histochemistry (IHC) Staining

Livers were processed as mentioned above (H&E staining section). Following dewaxing and antigen retrieval, liver sections were stained with a BOND Polymer Refine Detection Kit (DS9800, Leica, Munich, Germany) by BOND-III Automated IHC/ISH Stainer (Leica, Germany). Gp130 and Ki67 staining was examined by light microscopy. Positive cells were quantified from 3 randomly selected 200× field images of liver sections (saline: *n* = 5 mice/genotype; APAP: *n* = 6 mice/genotype) using the cell counter function in ImageJ software (National Institute of Health, Bethesda, MD, USA).

### 4.7. Statistical Analysis

Statistical analyses were performed using GraphPad Prism software (version 9, La Jolla, CA, USA). Simple two-tailed Student’s *t*-tests were used for experimental setups requiring testing of just two conditions. Comparison analysis for several conditions from two different groups were performed by 2-way ANOVA and corrected with Sidak’s multiple comparisons when the means were compared to each other. The criterion for statistical significance was *p* < 0.05.

## Figures and Tables

**Figure 1 ijms-23-07089-f001:**
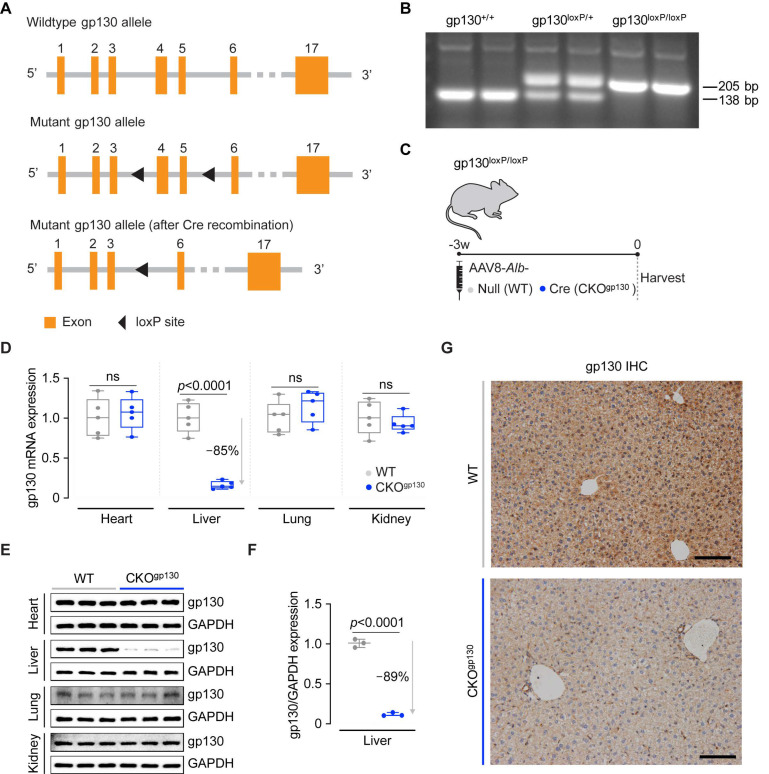
Generation and validation of hepatocyte-specific, gp130-deficient adult mice. (**A**) Schematic of loxP sites introduced into introns 3 and 5 of the gp130/*Il6st* locus used for the generation of gp130^loxP/loxP^ mice. (**B**) Representative genotyping of gp130^+/+^, gp130^loxP/+^, or gp130^loxP/loxP^ mice. (**C**) Schematic showing AAV8-induced, hepatocyte-specific gp130 knockout mice (CKO^gp130^) for data shown in (**D**–**G**). (**D**) gp130 mRNA (*n* = 5/group) and (**E**) gp130 protein expression (*n* = 3/group) in the heart, liver, lung, and kidney. (**F**) Densitometry analysis (*n* = 3/group) and (**G**) a representative IHC staining image of gp130 expression in mice liver (scale bars, 200 µm). (**D**,**F**) Data are shown in a box-and-whisker plot with median (middle line), 25th–75th percentiles (box), and minimum-maximum values (whiskers); 2-tailed Student’s *t*-test.

**Figure 2 ijms-23-07089-f002:**
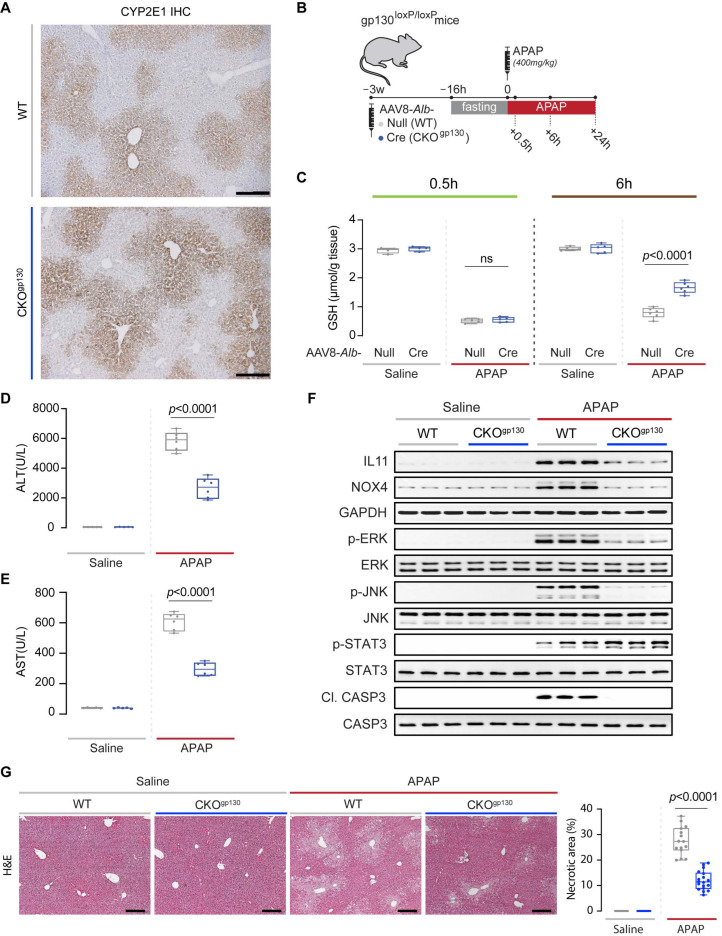
Hepatic CYP2E1 and glutathione levels and serum and molecular markers of liver injury in mice with hepatocyte-specific gp130 deletion and control mice. (**A**) Representative immunohistochemistry images (scale bars, 200 µm; representative dataset from *n* = 3 mice/group). (**B**) Schematic of APAP induction in wild-type (WT) and CKO^gp130^ mice: overnight-fasted mice were injected with saline or APAP (400 mg kg^−1^); livers and serum were collected either 0.5, 6, or 24 h post-APAP dosing for experiments shown in (**C**–**F**) and Figure 3. (**C**) Hepatic GSH concentrations at 0.5 and 6 h post-APAP. (**D**) ALT, (**E**) AST, and (**F**) Western blots of IL11, NOX4, GAPDH, p-ERK, ERK, p-JNK, JNK, p-STAT3, STAT3, Cl. CASP3, and CASP3 (*n* = 3/group) at 6 h post-APAP. (**G**) Representative H&E-stained liver images (scale bars, 200 µm) at 6 h post-APAP dosing and quantification of liver necrotic area. (**C**–**E**,**G**) Data are shown in a box-and-whisker plot with median (middle line), 25th–75th percentiles (box) and min-max values (whiskers), with a 2-way ANOVA with Sidak’s correction; 0.5 h time point: saline (*n* = 4 mice/genotype), APAP (*n* = 4 mice/genotype), 6: saline (*n* = 5 mice/genotype), APAP (*n* = 6 mice/genotype).

**Figure 3 ijms-23-07089-f003:**
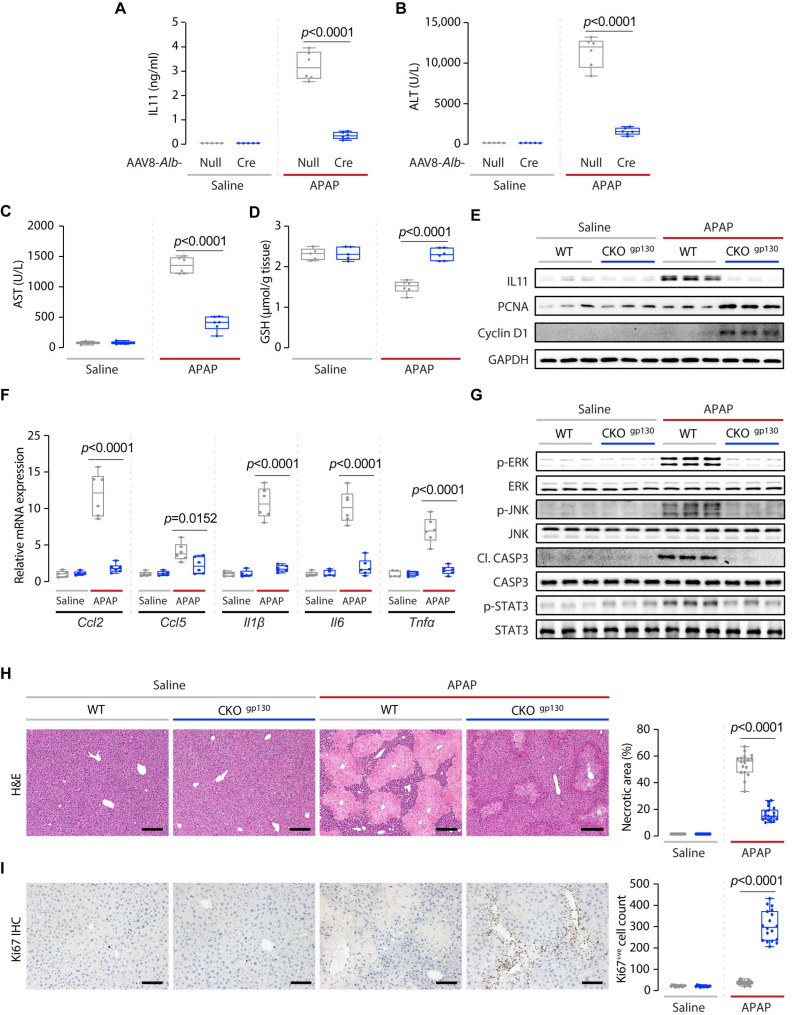
Hepatocyte-specific deletion of gp130 limits APAP-induced liver injury and promotes hepatic regeneration 24 h after injury. (**A**) Serum IL11 levels, (**B**) serum ALT levels, (**C**) serum AST levels, (**D**) hepatic GSH levels, (**E**) Western blots showing hepatic levels of IL11, PCNA, Cyclin D1, and GAPDH as internal contro (*n* = 3/group)l, (**F**) hepatic mRNA expression of pro-inflammatory markers (*Ccl2*, *Ccl5*, *Il1β*, *Il6*, *Tnfα*), (**G**) Western blots showing hepatic levels of p-ERK, ERK, p-JNK, JNK, Cl. CASP3, CASP3, p-STAT3, and STAT3 (*n* = 3/group), (**H**) representative H&E-stained liver images (scale bars, 200 µm) and quantification of liver necrotic area, (**I**) IHC staining of Ki67 in the livers of mice (scale bars, 200 µm) and quantification of Ki67^+ve^ cells for experiments shown in Figure 2B. (**A**–**D**,**F**,**H**,**I**) Saline (*n* = 5 mice/genotype), APAP (*n* = 6 mice/genotype); data are shown in a box-and-whisker plot with median (middle line), 25th–75th percentiles (box), and min-max values (whiskers), with a 2-way ANOVA with Sidak’s correction.

**Figure 4 ijms-23-07089-f004:**
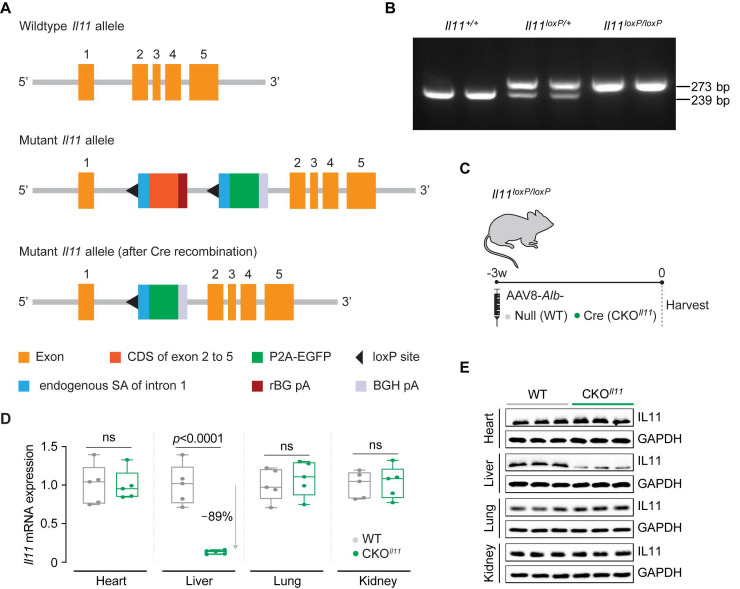
Generation and validation of hepatocyte-specific *Il11*-deficient adult mice. (**A**) Schematic design showing the deletion of exons 2–5 of the mouse *Il11* locus and the gene targeting strategy. (**B**) Representative genotyping of *Il11^+/+^*, *Il11^loxP/+^*, or *Il11^loxP/loxP^* mice. (**C**) Schematic showing AAV8-induced hepatocyte-specific *Il11* knockout mice (CKO*^Il11^*). (**D**) *Il11* mRNA (*n* = 5/group) and (**E**) IL11 protein (*n* = 3/group) expression in the heart, liver, lung, and kidney from WT and CKO*^Il11^* mice. (**D**) Data are shown in a box-and-whisker plot with median (middle line), 25th–75th percentiles (box), and minimum-maximum values (whiskers), with a 2-tailed Student’s *t*-test.

**Figure 5 ijms-23-07089-f005:**
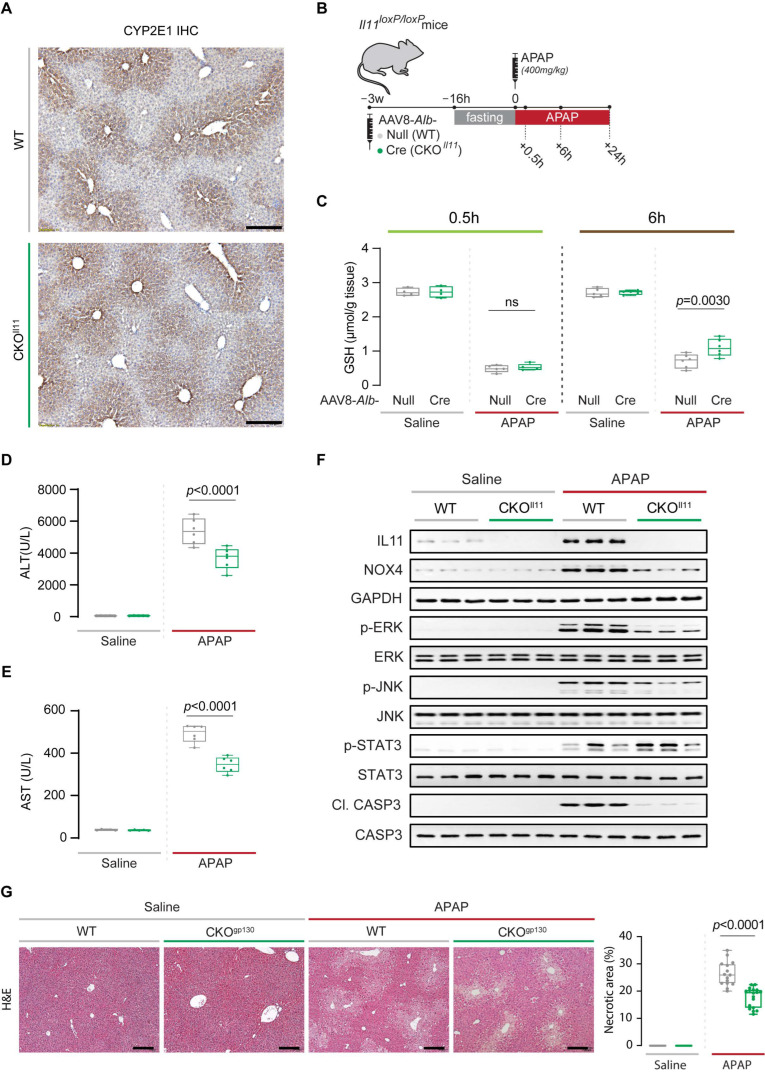
Hepatic CYP2E1 and glutathione levels and serum and molecular markers of liver injury in mice with hepatocyte-specific *Il11* deletion and control mice. (**A**) Representative immunohistochemistry images (scale bars, 200 µm; representative dataset from *n* = 3 mice/group). (**B**) Schematic of induction of APAP injury in wild-type (WT) and CKO*^Il11^* mice: overnight-fasted mice were intraperitoneally injected with saline or APAP (400 mg kg^−1^); livers and serum were collected either 0.5, 6, or 24 h following APAP administration for experiments shown in (**C**–**F**) and Figure 6. (**C**) Hepatic GSH concentrations at 0.5 and 6 h post-APAP. (**D**) ALT, (**E**) AST, and (**F**) Western blots of IL11, NOX4, GAPDH, p-ERK, ERK, p-JNK, JNK, Cl. CASP3, and CASP3 at 6 h post-APAP. (**G**) Representative H&E-stained liver images (scale bars, 200 µm) at 6 h post-APAP dosing quantification of the liver necrotic area. (**C**–**E**,**G**) Data are shown in a box-and-whisker plot with median (middle line), 25th–75th percentiles (box), and min-max values (whiskers), with a 2-way ANOVA with Sidak’s correction; 0.5 h time point: saline (*n* = 4 mice/genotype), APAP (*n* = 4 mice/genotype), 6 h time point: saline (*n* = 5 mice/genotype), APAP (*n* = 6 mice/genotype).

**Figure 6 ijms-23-07089-f006:**
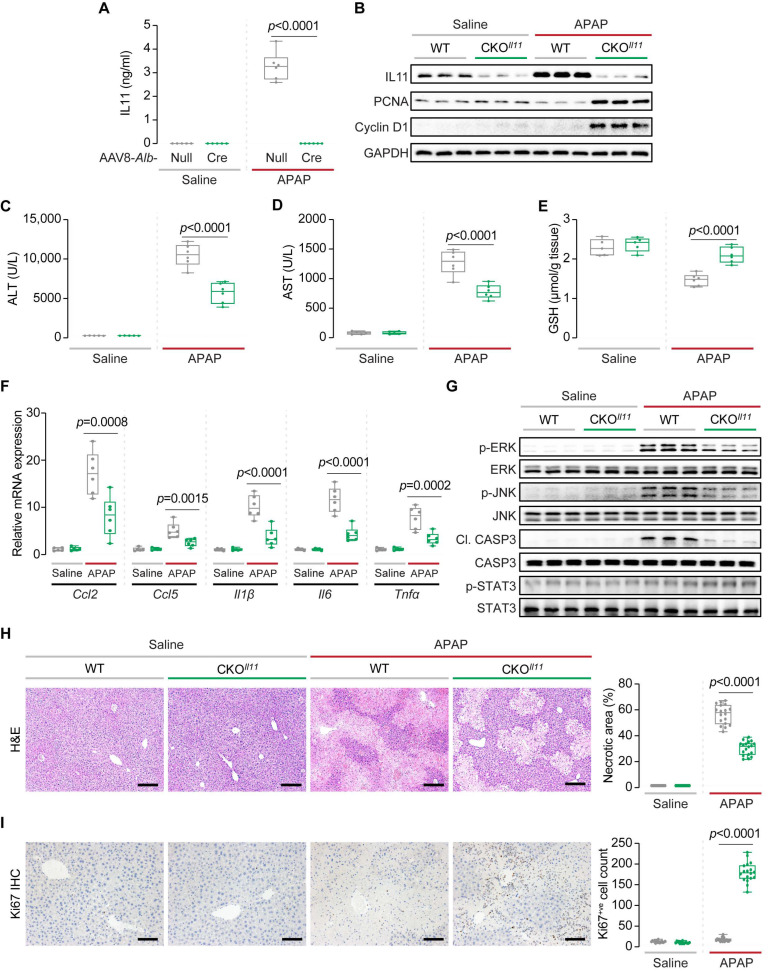
Hepatocyte-specific deletion of *Il11* reduces APAP-induced liver injury and promotes hepatic regeneration 24 h after injury. (**A**) Serum IL11 levels, (**B**) Western blots showing hepatic levels of IL11, PCNA, Cyclin D1, and GAPDH as the internal control (*n* = 3/group), (**C**) serum ALT levels, (**D**) serum AST levels, (**E**) hepatic GSH levels, (**F**) hepatic mRNA expression of pro-inflammatory markers (*Ccl2*, *Ccl5*, *Il1β*, *Il6*, *Tnfα*), (**G**) Western blots showing hepatic levels of p-ERK, ERK, p-JNK, JNK, Cl. CASP3, CASP3, p-STAT3, and STAT3 (*n* = 3/group), (**H**) representative H&E-stained liver images (scale bars, 200 µm) and quantification of the liver necrotic area, (**I**) IHC staining of Ki67 in the livers of mice (scale bars, 200 µm) and quantification of Ki67^+ve^ cells for experiments shown in Figure 5B. (**A**,**C**–**F**,**H**,**I**) Saline (*n* = 5 mice/genotype), APAP (*n* = 6 mice/genotype); data are shown in a box-and-whisker plot with median (middle line), 25th–75th percentiles (box), and min-max values (whiskers), with a 2-way ANOVA with Sidak’s correction.

**Figure 7 ijms-23-07089-f007:**
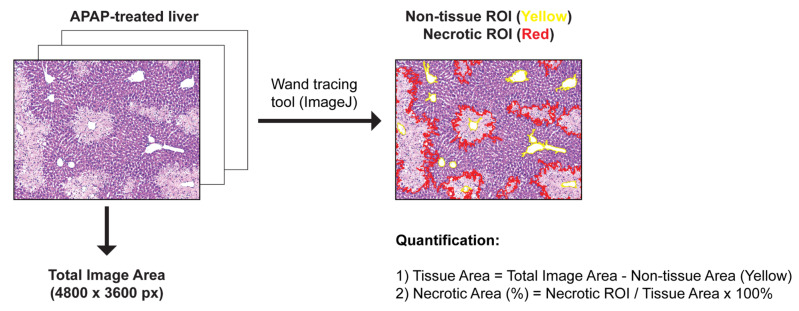
Exemplifies the quantification analysis.

## Data Availability

The original contributions presented in the study are included in the article/Appendix A; further inquiries can be directed to the corresponding authors.

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
