# Peer review of "Hepatocyte Specific gp130 Signalling Underlies APAP Induced Liver Injury"

_ijms, 2022, doi:10.3390/ijms23137089_

Round 1

Reviewer 1 Report

In their study ”Hepatocyte specific gp130 signalling underlies APAP induced liver injury” Dong et al. show, using two newly generated conditional KO mouse models for gp130 and IL11, that liver damage caused by an acetaminophen overdose can be ameliorated by reducing IL11 signalling via gp130 in hepatocytes.

Overall the study is well designed and thoroughly executed. The authors do make a convincing argument that hepatocytes are the major source of IL-11 secretion upon APAP-induced liver injury. However, the study is lacking in some aspects as the described results are mostly observational and no mechanistic explanation is pursued.

Major comment:

1.       One very important aspect of APAP-induced liver injury is not at all investigated or discussed, though it might be quite relevant for the observed phenotypes – the role of senescence in liver regeneration after acute liver injury. IL-11 is considered one of the cytokines associated with the senescence-associated secretory phenotype (SASP). Furthermore, IL1beta, IL6 and TNFa, all found to be downregulated in both CKOgp130 and CKOIL11 mice, are part of the SASP. Therefore, this potential angle needs to be investigated and either confirmed or excluded as a potential mechanism for the observed phenotype

2.       It would be easier to interpret the findings if the data weren’t shown as individual separate time points but rather as a time course, which would make it easier for the reader to appreciate the differences in eg. Peak damage in the models. Furthermore, it would be equally of interest to the reader to see H&E stainings of the earlier time points and the APAP-induced damage as well as mRNA expression of the investigated pro-inflammatory markers (or SASP markers?).

Minor comments:

1.       L85: Reference 24 (Ho et al. Optimized Adeno-Associated Virus 8 Produces 529 Hepatocyte-Specific Cre-Mediated Recombination without Toxicity or Affecting Liver Regeneration. Am. J. Physiol. Gastrointest. 530 Liver Physiol. 2008) describes AAV-MUP-iCre rather than the AAV-Alb-iCre used in this study. A reference showing specificity of AAV-Alb-iCre is needed instead.

2.       Figure 3H: the H&E image of the CKOgp130 mouse does not seem to be representative of the quantification of the necrotic area. Can the authors choose a more representative image?

Reviewer 2 Report

I read this paper and I think that:

This study is a very interesting, and it is considered a very necessary study for the clinical success of hepatotoxic protection

In this article, authors well summarized the role of IL11 and gp130 in APAP liver injury.

However, there are a few questions.

1.     In introduction, authors mentioned that three questions below: 

(1) Is gp130 signaling in hepatocytes protective or injurious in the context of APAP-induced liver injury.

(2) are hepatocytes a major source of IL11 secretion following APAP injury.  

(3) is endogenous IL11 activity in APAP injured hepatocytes adaptive or maladaptive?

               Please answer these questions in discussion part.

2.    This study showed that liver injury was reduced in hepatocyte-specific Il11-deficient mice.

How does IL11 overexpression affect APAP-induced liver injury?

3.     IL-6 and IL-11 are closely related to inflammation. 

How dose hepatocyte-specific deletion of gp130 and IL-11 affect liver inflammation?

Reviewer 3 Report

The manuscript by Dong et al describes the roles of hepatocyte specific gp130 signaling on the emergence of APAP-induced hepatotoxicities using conditional knockout mice of either gp130 or IL11. Authors showed the involvement of autocrine self-amplification of loop of IL11secretion in APAP hepatotoxicities. The contents are advanced and interesting and deserve publication in IJMS. Before acceptance, it would be desirable if following items were improved.

1. Line 2 (Title): Please use a formal name of APAP such as N-acetyl-p-aminophenol or Paracetamol for rapid understanding of outsiders.

2. Line 30 (Introduction): It may be more appropriate to use the word “conservative” than “ancient”.

3. Line 37 (Introduction): It may be more appropriate to use the word “some” than “old”, because reference 4 is not so old.

4. Line 108-111 (Results): Please add more explanatory remark to Figure 2A. In addition, it would be informative to show quantitative comparison of CYP2E1 somewhere (brief mentioning in the text is OK), if possible.

5. Figure 4D: Please add the indication, “Null” and “Cre” on abscissa because there is no precedent presentation in Figure 4.

6. Line 335-341 (Conclusion section in Discussion): Please mention the conclusion of each working hypothesis proposed in the last part of Introduction (Line 66 to 69) one by one to clarify the significance of CKOgp130 mice data.
